# Deep Learning with Open Data for Desert Road Mapping

**Christopher Stewart [1],\* , Michele Lazzarini [2], Adrian Luna [2] and Sergio Albani [2]**

[1]   European Space Agency (ESA), Earth Observation Programmes, Future Systems Department, 00044 Frascati, Italy

[2]   European Union Satellite Centre (SatCen), 28850 Madrid, Spain; Michele.Lazzarini@satcen.europa.eu (M.L.); Adrian.Luna@satcen.europa.eu (A.L.); Sergio.Albani@satcen.europa.eu (S.A.)

\*   Correspondence: chris.stewart@esa.int

**Abstract:** The availability of free and open data from Earth observation programmes such as Copernicus, and from collaborative projects such as Open Street Map (OSM), enables low cost artificial intelligence (AI) based monitoring applications. This creates opportunities, particularly in developing countries with scarce economic resources, for large–scale monitoring in remote regions. A significant portion of Earth's surface comprises desert dune fields, where shifting sand affects infrastructure and hinders movement. A robust, cost–effective and scalable methodology is proposed for road detection and monitoring in regions covered by desert sand. The technique uses Copernicus Sentinel–1 synthetic aperture radar (SAR) satellite data as an input to a deep learning model based on the U–Net architecture for image segmentation. OSM data is used for model training. The method comprises two steps: The first involves processing time series of Sentinel–1 SAR interferometric wide swath (IW) acquisitions in the same geometry to produce multitemporal backscatter and coherence averages. These are divided into patches and matched with masks of OSM roads to form the training data, the quantity of which is increased through data augmentation. The second step includes the U–Net deep learning workflow. The methodology has been applied to three different dune fields in Africa and Asia. A performance evaluation through the calculation of the Jaccard similarity coefficient was carried out for each area, and ranges from 84% to 89% for the best available input. The rank distance, calculated from the completeness and correctness percentages, was also calculated and ranged from 75% to 80%. Over all areas there are more missed detections than false positives. In some cases, this was due to mixed infrastructure in the same resolution cell of the input SAR data. Drift sand and dune migration covering infrastructure is a concern in many desert regions, and broken segments in the resulting road detections are sometimes due to sand burial. The results also show that, in most cases, the Sentinel–1 vertical transmit–vertical receive (VV) backscatter averages alone constitute the best input to the U–Net model. The detection and monitoring of roads in desert areas are key concerns, particularly given a growing population increasingly on the move.

**Keywords:** synthetic aperture radar; SAR; Sentinel–1; Open Street Map; deep learning; U–Net; desert; road; infrastructure; mapping; monitoring

## 1. Introduction

The mapping and monitoring of roads in desert regions are key concerns. Population growth and an increase in the development of urban centres have led to a corresponding expansion of transportation networks [1,2]. These networks are constantly evolving [1,3]. An awareness of the location and state of road systems is important to help monitor human activity and to identify any maintenance that may be required for the infrastructure. In many desert regions, roads and tracks are

used for illicit activities, such as smuggling [4]. Sand drift and dune migration can rapidly bury roads, thus necessitating intervention [5–7].

Ground techniques used for surveying and monitoring road networks are expensive and time consuming [2]. This is especially true for desert regions, given the extensive areas involved, the often inhospitable landscapes, and, in some cases, the political instability [8,9]. Remote sensing techniques have the ability to acquire information over large areas simultaneously, at frequent intervals, and at a low cost [10,11]. The application of emerging technologies, such as big data, cloud computing, interoperable platforms and artificial intelligence (AI), have opened new scenarios in different geospatial domains [12], such as the monitoring of critical infrastructure, i.e., roads.

Previously developed algorithms to automatically extract road features using techniques such as classification, segmentation, edge and line detection and mathematical morphology are summarised in a number of review papers, such as [13–16]. Since deep convolutional neural networks proved their effectiveness in the 2012 ImageNet Large Scale Visual Recognition Challenge (ILSVRC), deep learning has significantly gathered pace. Among the first to apply deep learning for road extraction were Mnih and Hinton [17]. Saito and his colleagues later achieved even better results with convolutional neural networks (CNNs) [18]. Techniques using CNNs are now considered to be standard for image segmentation [19] with many studies proposing different CNN architectures for road detection and monitoring, e.g., [1,3,20–24]. This is a fast evolving domain, and new research is regularly published on architectures and methods to address some of the limitations of CNNs. These include, for example, the significant computing and memory requirements [25], the fact that much training data is often needed, and the difficulty in adapting models to varying conditions [26]. A particularly effective CNN model for semantic segmentation is the U–Net architecture. Devised by Ronneberger and his colleagues for medical image segmentation [27], U–Net has become a standard technique for semantic segmentation in many applications since it won the IEEE International Symposium on Biomedical Imaging (ISBI) cell tracking challenge in 2015 by a large margin. The popularity of this architecture, which consists of a contracting path to capture the context and a symmetric expanding path that enables precise localisation, is due partly to its speed, and its ability to be trained end–to–end with very few images [27]. Many have applied variations of U–Net for road detection, e.g., [2,3,22–24], the majority basing their models on dedicated benchmark datasets of optical images for road identification, such as the Massachusetts roads data, created by Mihn and Hinton [17].

Most remote sensing based techniques for road detection and monitoring have relied on very high resolution (VHR) optical data [13]. However, in desert regions the spectral signatures of roads are often similar to the surrounding landscape, making them difficult to distinguish. Synthetic aperture radar (SAR) data has characteristics which make it efficient in the retrieval of roads in desert regions [9,28]. These include the sensitivity of the radar to surface roughness and the relative permittivity of targets, and the fact that SAR is a coherent system [29]. Dry sand usually has a very low relative permittivity and is therefore not a high reflector of microwave radiation. Sand covered areas are thus usually characterised by a very low SAR backscatter. Roads on the other hand may display a very different type of backscatter, which can contrast highly with the surrounding sand, even if the roads are significantly narrower than the SAR resolution cell [9]. These characteristics can be exploited to retrieve roads from SAR amplitude data. SAR coherence can also help to detect roads in desert regions. The low relative permittivity of dry sand causes the transmission of the microwave SAR signal into the sand volume [30,31]. Coherence is rapidly lost in such areas due to volume decorrelation [32]. This low coherence may contrast with the higher coherence of roads, often made from materials with a higher relative permittivity, such as asphalt, tarmac, or gravel, which therefore are not affected by volume decorrelation.

Some studies nonetheless have demonstrated methodologies for road detection and monitoring using SAR data. A good review of many of these is provided by [14]. More recently, a few studies have successfully applied deep learning techniques for SAR based road detection, e.g., [1,2,21], but these have mainly focused on relatively small, local areas, in developed landscapes, where good ground

truth and training data have been available. Some have also used SAR for detecting roads and tracks in desert regions, e.g., Abdelfattah and his colleagues proposed a semi–automatic technique for SAR based road detection over a local area in the Tunisian–Libyan border [4], but again, this was applied to a specific area, and was not fully automatic.

Robust methodologies are required for operational road detection and monitoring in desert regions over large areas without the need to acquire expensive reference data. Many desert areas are situated in developing countries, such as in North Africa, where accurate and abundant training data are not available, and budgets for infrastructure surveying are low.

The work presented in this paper aims to demonstrate a methodology for road detection and monitoring in desert regions, using free input and reference data that can be scaled to desert regions globally. This approach takes input SAR data from the free and open Copernicus Sentinel–1 satellite constellation over the area to be surveyed. The input data comprises both the amplitude and coherence averages from a time series of around two and a half months acquired in the same geometry (around seven scenes). The time series average contributes to removing image speckle and improves the model performance. The reference data on the other hand includes freely available Open Street Map (OSM) data. The combined use of OSM and Earth observation (EO) data in semantic segmentation has been much discussed, e.g., [33–35], but in most cases it has been used either with very high resolution (VHR) EO data, or for general classes with much less class imbalance than the road, no–road distinction. Roads are then extracted using a version of U–Net. With its architecture consisting of a contracting path to capture the context and a symmetric expanding path that enables precise localisation, U–Net has the well–known advantages that it can be trained end–to–end with very few images, and is fast [27]. This makes it suitable for cases where abundant, high quality reference data may not be available. One of the many versions of this architecture adapted to Earth observation data includes one proposed by Jerin Paul that was previously applied successfully to VHR optical data [36]. This was the version adopted in this methodology. Despite the fact that it was developed for use with optical data, it performed well on SAR based inputs with similar class imbalance. This U–Net model is trained with SAR amplitude and coherence averages, with OSM reference masks, for each desert region. The model is then applied to detect roads in each of the desert areas for which is was trained.

The method proposed here for SAR based deep learning segmentation, trained on OSM data, has been applied to a number of test areas in various deserts in Africa and Asia. The high accuracy of the results suggests that a robust methodology involving the use of freely available input and reference data could potentially be used for operational road network mapping and monitoring.

This study has been carried out in the framework of a joint collaboration between the European Space Agency (ESA) Φ–Lab, and the European Union Satellite Centre (SatCen) Research, Technology Development and Innovation (RTDI) Unit. The ESA Φ–Lab carries out research in transformative technologies that may impact the future of EO. The SatCen RTDI Unit provides new solutions supporting the operational needs of the SatCen and its stakeholders by looking at the whole EO data lifecycle.

## 2. Materials and Methods

This section presents the methodology for road detection and monitoring using free and open data. The process can be divided into two steps: The first is a SAR pre–processing step, to obtain temporal averages of the calibrated backscatter amplitude and consecutive coherence for each time series, over each area. The second is the deep learning workflow. In this second step, the input SAR layers are divided into adjacent, non–overlapping patches of 256 × 256 pixels. Each patch is matched with a corresponding mask of the same resolution and dimension showing the location of any OSM roads. In these masks, pixels coinciding with roads have a value of 1, while all other pixels have a value of 0. All SAR patches, which included OSM roads in their corresponding masks, were used to train the U–Net model, initiated with random weights, using the OSM data as a reference. Subsequently, the model was applied to all patches in each area of interest (AOI) to extract the roads not included

in the OSM dataset. The AOIs included three areas in three different desert environments in Africa and Asia, each the size of one Sentinel–1 IW scene (250 × 170 km).

While the OSM was used as the reference for model training, a more precise dataset was needed for an accuracy assessment. The reference masks only recorded roads present in the OSM dataset. The possibility existed that roads were present in the coverage of any given reference mask patch, but not included in the OSM. Moreover, due to the varying quality of the OSM and the varying width of roads, precise overlap between the model detected roads and OSM reference masks was difficult to achieve. To maintain automation and ensure the scalability of the method, there was no manual editing of these patches. Nonetheless, for the purpose of model training, the procedure to use the OSM as the reference worked well. For a reliable accuracy assessment however, a more rigorous technique was adopted: a subset area was randomly selected in each desert region in which all roads were manually digitised. These data were then used as the reference for a more precise accuracy assessment.

## 2.1. Areas of Interest (AOIs)

Three AOIs in different types of sand covered deserts were chosen to apply the method. These include most of the North Sinai Desert of Egypt, a large part of the Grand Erg Oriental in the Algerian and Tunisian Sahara, and the central part of the Taklimakan Desert of China (see Figure 1). The size of each of these three areas corresponds to the extent of one Sentinel–1 interferometric wide swath (IW) scene: 250 × 170 km, covering an area of 47,500 km$^2$ in each desert region. They were chosen for their geographic and morphological variety, each having very different sand dune forms and local conditions.

## Map of AOIs

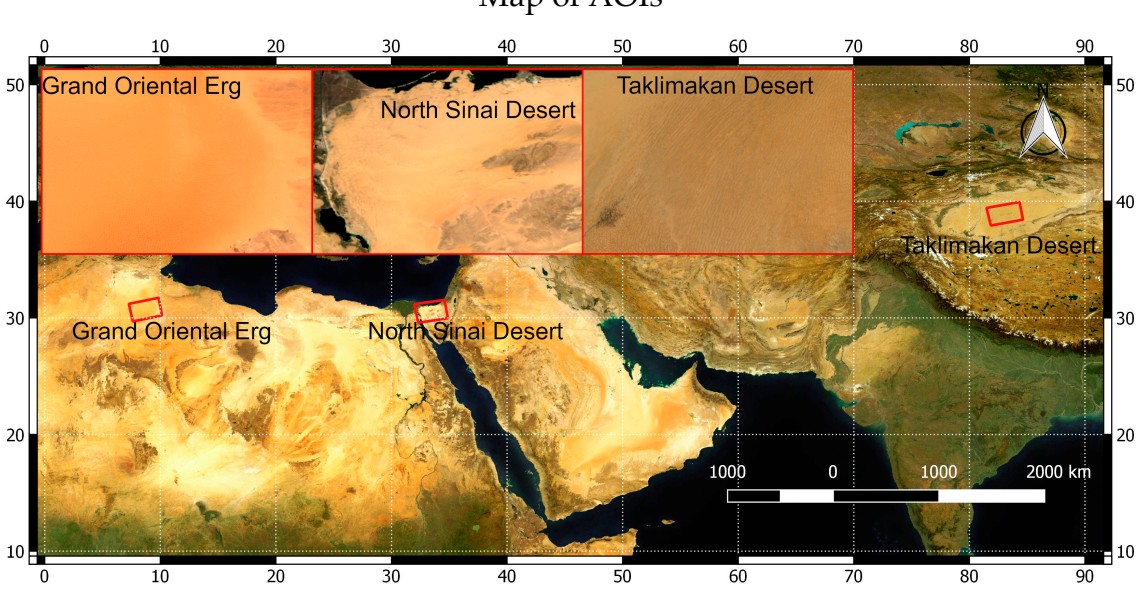

**Figure 1.** Map showing the location of areas of interest (AOIs) on an ENVISAT MERIS true colour mosaic, in a geographic latitude, longitude map system, World Geodetic System 1984 (WGS84) datum. Insets show a close–up of the AOIs. Each AOI and inset has the dimension of one Sentinel–1 IW footprint (250 km East–West, 170 km North–South). Credits: CHELYS srl for the world map and the European Space Agency (ESA) GlobCover for insets.

The North Sinai Desert, in the north of the Sinai Peninsula, is composed mainly of aeolian sand dune fields and interdune areas. The sand dunes include barchan, seif or longitudinal linear dunes trending east–west, transverse and star dunes [37]. Linear dunes are the main aeolian form in North Sinai [5]. The climate of the study area is arid. The average annual rainfall is about 140 mm at El Arish [38], but drops in the south, where it does not exceed 28 mm per year [5].

The Grand Erg Oriental is a sand dune field in the Sahara desert, mainly in Algeria, but with its north–eastern edge in Tunisia. It is characterised by four large–scale dune pattern types with gradual transitions between them. These include large, branching linear dunes; small and widely spaced star and dome dunes; a network type created mostly from crescentic dunes; and large, closely spaced star dunes [39]. The average annual rainfall does not exceed 70 mm [40].

The Taklimakan Desert is the world's second–largest shifting sand desert, located in China, in the rain shadow of the Tibetan Plateau [41]. Three types of sand dunes exist in the Taklimakan Desert: compound, complex crescent dunes and crescent chains; compound dome dunes; and compound, complex linear dunes [42]. The mean annual precipitation varies between 22 and 70 mm [43].

### 2.2. SAR and OSM Data

To achieve the objective of demonstrating a robust and cost–effective methodology that can be applied globally, it was decided to exploit the Copernicus Sentinel–1 archive. The Sentinel–1 data are acquired at regular intervals worldwide and are available under a free and open access policy [44]. Over each of the AOIs, a time series was obtained of 7 images acquired every 12 days over an approximately two–and–a–half–month period (June/July to August/September 2019). The images were all interferometric wide swath (IW), all in ascending geometry and dual polarisation: vertical transmit–vertical receive, and vertical transmit–horizontal receive (VV and VH, respectively). In order to explore the use of both amplitude and coherence in road detection, the time series over each area was obtained in both ground range detected (GRD) and single look complex (SLC) formats. The spatial resolution of the Sentinel–1 IW data is approximately $20 \times 20$ metres for the GRD and $5 \times 20$ metres for the SLC. The pixel spacing of the GRD data is $10 \times 10$ metres. Table 1 shows the details of the Sentinel–1 data used in each AOI. All GPT graphs and bash scripts are available on Github (A Github repository has been created which contains all scripts that were used in this research, including the Bash files and GPT graphs for the Sentinel–1 data processing, and the Python code for the deep learning workflow available in a Jupyter Notebook. In this repository are also results in a shapefile format of the road detections over each of the AOIs. Supplementary data—Available online: https://github.com/ESA-PhiLab/infrastructure) [45].

**Table 1.** Details of the Sentinel–1 time series used in each of the AOIs.

| AOI | Orbit | Polarisation | Time Series Length |
|---|---|---|---|
| North Sinai Desert | Ascending | VV/VH | 7 scenes acquired from 4 July to 14 September 2019 |
| Grand Erg Oriental | Ascending | VV/VH | 7 scenes acquired from 4 June to 15 August 2019 |
| Taklimakan Desert | Ascending | VV/VH | 7 scenes acquired from 11 June to 22 August 2019 |

OSM data, including all roads, was downloaded at continental scale. From the original XML formatted osm files, they were converted to vector shapefiles; subsequently, the OSM data were subset for each AOI and attribute fields were reduced to the sole roads identification, in order to limit the file size.

### 2.3. SAR Pre–Processing

Given that roads in desert areas can be distinguished in both SAR amplitude and coherence, it was decided to include both as inputs to the U–Net model. A virtual machine (VM), with Ubuntu as the operating system, was used for the Sentinel–1 pre–processing. This VM was connected to the CreoDIAS cloud environment, containing archive Sentinel–1 data. Processing was carried out automatically on the cloud using the command line graph processing tool (GPT) of the open source ESA Sentinel application platform (SNAP) software. Two GPT graphs, including all steps of each of

the SLC and GRD processing chains, were applied in batch to the time series of data over each area using Linux bash scripts.

### 2.3.1. Amplitude Processing

For the amplitude processing, each Sentinel–1 scene, in a GRD format, was calibrated to $\sigma^0$ backscatter. The calibrated data was then terrain corrected to the European Petroleum Survey Group (EPSG) 4326 map system, i.e., geographic latitude and longitude with the World Geodetic System 1984 (WGS84) datum. The topographic distortion was corrected with the aid of the shuttle radar topography mission (SRTM) 3 s global digital elevation model (DEM). The output pixel spacing was 10 m. The stack of calibrated and terrain corrected scenes was then co–registered using cross correlation. The co–registered stack was averaged into one scene to reduce speckle. This average was finally converted from the linear backscatter scale to logarithmic decibel, to improve visualisation and facilitate further pre–processing during the deep learning workflow.

Some very good multitemporal speckle filters exist that preserve the spatial resolution while also keeping the temporal backscatter differences, such as the De Grandi speckle filter [46]. This allows for the monitoring of temporal intervals of less than the length of the time series. However, the emphasis of the study was to demonstrate a robust methodology that uses open data and tools. The most effective way to sufficiently reduce speckle while completely preserving the spatial resolution using the tools available was to average the data. Figure 2 shows the steps of the processing chain applied automatically to the time series of the Sentinel–1 GRD data.

## Sentinel–1 intensity processing chain

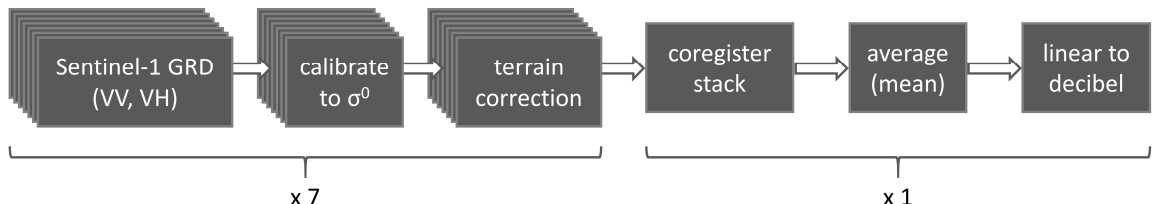

**Figure 2.** The intensity processing chain applied automatically to the Sentinel–1 data in the CreoDIAS cloud environment. The numbers below show how the time series of seven images are reduced to one layer for each area in which the model is applied.

### 2.3.2. Coherence Processing

For the coherence processing, the interferometric coherence was calculated for each consecutively acquired Sentinel–1 image pair in SLC format. For a time series of seven images therefore, six coherence images were obtained. These were then averaged to reduce clutter and better distinguish roads from the surrounding sand.

The steps in the coherence generation workflow began with the calculation of precise orbits. The three subswaths of each pair were then split to enable back–geocoding, coherence generation and TOPSAR–debursting to be carried out per subswath. These were then merged, before the coherence for each full scene pair was terrain corrected, to the same map system as used for the amplitude data processing. All terrain corrected coherences were co–registered, using a cross correlation, and averaged by taking the mean coherence for each pixel. The coherence average was finally resampled to the same pixel spacing as the amplitude average for each area, to enable the simultaneous (amplitude and coherence) input to the U–Net model. Figure 3 shows the main steps of the processing chain applied automatically to the time series of Sentinel–1 SLC data.

Sentinel–1 coherence processing chain

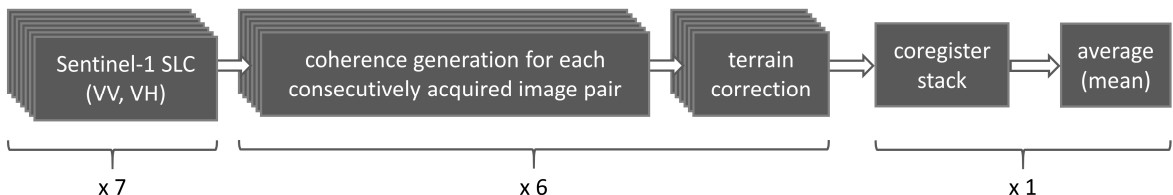

**Figure 3.** The main steps of the coherence processing chain applied automatically to the Sentinel–1 data in the CreoDIAS cloud environment. The numbers below show how the time series of seven images are reduced to one layer for each area in which the model is applied.

## 2.4. Deep Learning Workflow

The output of the Sentinel–1 pre–processing included three separate backscatter and coherence averages for each of the three desert areas, each covering the extent of one Sentinel–1 IW footprint ($250 \times 170$ km). These, together with the OSM ancillary data, comprised the input to the second part of the methodology, which included the deep learning workflow. This second part was developed in a sandbox environment for AI projects, called Sandy, belonging to the ESA Advanced Concepts Team. It includes 10 NVIDIA GTX1080 Ti graphics processing units (GPUs), suitable for training deep neural networks, although only one GPU was necessary for the model training. The deep learning workflow was implemented in Python 3, with Keras and Tensorflow. This Jupyter Notebook is available on Github (Supplementary data—Available online: https://github.com/ESA-PhiLab/infrastructure) [45].

### 2.4.1. OSM and SAR Data Preparation for Deep Learning

For each backscatter and coherence scene average, a corresponding mask was produced of the same extent and spatial resolution, in which pixels overlapping with OSM roads have a value of 1, and all other pixels have a value of 0. These masks were created by converting the OSM road vectors to raster.

Each SAR derived scene average and corresponding OSM mask were split into $256 \times 256$ adjacent, non–overlapping patches, and the SAR patches were normalised to contain pixel values from 0 to 1.

### 2.4.2. Data Augmentation

Those SAR and mask patches in which OSM roads were present, comprised the samples to train the U–Net model. The number of such patches per AOI varied from around 400 to 800. Data augmentation was therefore used to increase the number of training samples. The data augmentation included a random rotation through a 360–degree range. It also included a horizontal and vertical flip. These random transformations were chosen as they preserve exactly the width and backscatter characteristics of roads and surrounding features, while changing only their orientation. This was considered the best choice given that roads may feasibly have any orientation, while it is uncertain as to how much their width and backscatter properties may vary. To fill gaps in the image patches following transformation, a reflect mode was selected, i.e., any blank areas, e.g., corners of patches following rotation, are converted to the mirror image of the same number of non–blank pixels on the other side of the dividing line. This was the best possible fill mode given that the lines are not broken, but continue as would be expected with the roads. Figure 4 shows an example of random data augmentation for an input SAR patch and its corresponding OSM derived reference mask.

The data augmentation was implemented through an instance of the image data generator class of the Keras library for Python 3.7.

**Data augmentation.**

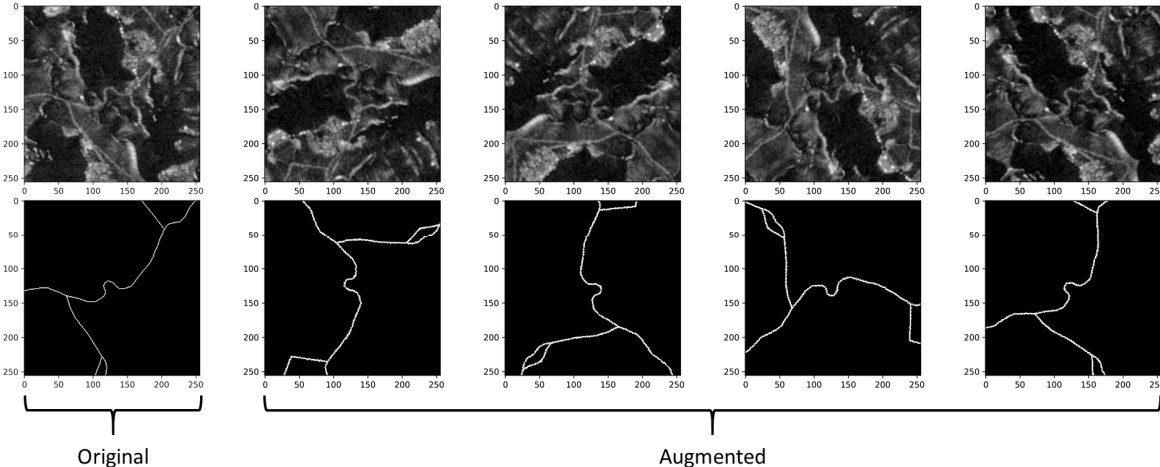

**Figure 4.** An example of data augmentation for an input synthetic aperture radar (SAR) patch (upper leftmost) and its corresponding reference mask (lower leftmost). Note how the augmented versions continue segments without breaking lines, using the "reflect" mode to fill gaps following rotation. Contains modified Copernicus Sentinel–1 data 2020.

### 2.4.3. U–Net Model

The deep learning model for image segmentation that was chosen is the modified U–Net architecture proposed by Jerin Paul [36]. This architecture has 58 layers, divided into a downsampling encoder and upsampling decoder parts, which are connected via skip connections. The convolution layers are all $3 \times 3$, with exponential linear units as the activation and He normal initialiser. The only exception to this is the last output layer, which is a $1 \times 1$ convolution layer with sigmoid activation. In between the convolution layers, batch normalisation, max pooling and data dropout layers were included. The data dropout layers applied a dropout rate varying from 0.1 to 0.3. The total number of parameters were 1,946,993. The trainable parameters were 1,944,049. All models were initiated with random weights.

The input to the network included up to three layers of average backscatter intensity in VV and VH, and average coherence. The models returned segmented images for each input patch, with pixels ranging in value from 0 to 1. Values closer to 0 have a high prediction probability of belonging to the class of non–roads, while values closer to 1 have a high probability of being a road.

### 2.4.4. Loss Function and Performance Metric

Road detection in desert regions is an unbalanced segmentation task, since in any given scene there are many more pixels falling into the non–road category than into the road category. The loss function applied during model training, and the accuracy metric to assess the performance of the model, need to take into account this class imbalance. The loss function that was applied in this case is the soft Dice loss. Based on the Dice coefficient, the soft Dice loss normalises for class imbalance. For the accuracy metric, it was decided to use the Jaccard index, also known as the Jaccard similarity coefficient, or intersection over union (IoU), which likewise considers class imbalance [47].

The formulae for soft Dice loss and the Jaccard index for a model predicted class ($A$) and a known class ($B$) are the following:

$$\text{Soft Dice Loss} = 1 - \frac{2|A \cap B|}{|A| + |B|} \tag{1}$$

$$\text{Jaccard index} = \frac{|A \cap B|}{|A \cup B|} \tag{2}$$

### 2.4.5. Hyperparameters and Model Training

Different approaches were attempted for the model training. One approach was to include the available training data from all areas with the aim of training one model applicable to every sand covered desert landscape with characteristics similar to those of the test areas. It was soon apparent that this was not feasible, due mainly to the greatly varying sand dune forms between different desert environments. This led to systematic false detections and almost no positive road detections over any area. Another approach was to choose one model, which would be trained for each specific desert region, with the available training data from that area. With this approach, even with much less training data, the model performed much better.

In addition to experimenting with the geographic coverage, different types of Sentinel–1 input were tested. Various combinations of the VV and VH backscatter and coherence were included as inputs to the model, from individual bands, to combinations of two, or all three.

The model hyperparameters are listed in Table 2. After testing different values for each parameter, these provided the best results for all the regions in which the method was applied, and with all the options for the SAR inputs. The only area specific parameter to be adjusted is the number of steps per epoch, which depends on the amount of training samples over a given region. Apart from the steps per epoch, these are the same hyperparameters as is in the model of Jerin Paul [36].

**Table 2.** Model hyperparameters.

| Model Hyperparameters | |
|---|---|
| Epochs | 100 |
| Steps per epoch | Training samples/Batch size, (minimum of 50) |
| Learning rate | 0.0001 |
| Batch size | 16 |

After randomly shuffling all samples, 10 percent were set aside for validation to assess the performance of the model during training, and another 10 percent for testing. After a review of this, a second round of training was carried out using all available data. Given the incompleteness of the OSM data in any given area, and the poor overlap between the OSM and detected roads discussed above, a more reliable accuracy assessment was carried out with the test data comprising manually digitised roads over subset areas. This is described in Section 2.5 below.

### 2.4.6. Post–Processing and Final Map Generation

Over each area, having trained the model with the image patches containing the available OSM data, the model was applied to predict the presence of roads in all patches. The pixels in the resulting segmented patches ranged from 0 to 1. Values closer to 0 represented a high probability of belonging to the no–road class, while those closer to 1 were considered likely to be roads. To create binary masks, all pixels with a value of less than 0.5 were converted to 0, and those greater than or equal to 0.5 were converted to 1. The patches were then put together to reconstruct the original image scene. Finally, the resulting single raster mask was converted to a vector layer containing all the predicted roads as polygon vectors in one shapefile.

### 2.5. Performance Evaluation

A performance evaluation of the methodology was carried out by manually digitising all the roads in subset areas within each AOI, and comparing these with the model detections through the calculation of the Jaccard similarity coefficient and the rank distance. The rank distance in turn is a measure which combines the completeness (percentage of reference data covered by model detections), and correctness (percentage of model detections covered by reference data) [48]. A performance evaluation with

manually digitised reference data was necessary given the following limitations of using the OSM data as a reference.

1.  The quality of the OSM data varied, in some cases road meanders were represented by straight lines (see Figure 5). This caused a misregistration between actual and OSM roads. In these cases, the U–Net model could still associate the same roads in the SAR and OSM masks by downscaling through layers of convolution filters, but for the automatically calculated IoU, the misregistration could mean no overlap (unless large buffers are used), and hence the SAR and OSM roads would not match.

2.  In addition to the misregistration between the modelled and reference data, another challenge was the missing roads in the OSM data. The intention of the methodology was to detect roads unrecorded in the OSM dataset, but in the same geographic area. The chances of roads being missed in many of the reference OSM mask patches was therefore high. In the interest of demonstrating a robust and scalable methodology, manual editing to improve the mask patches was avoided.

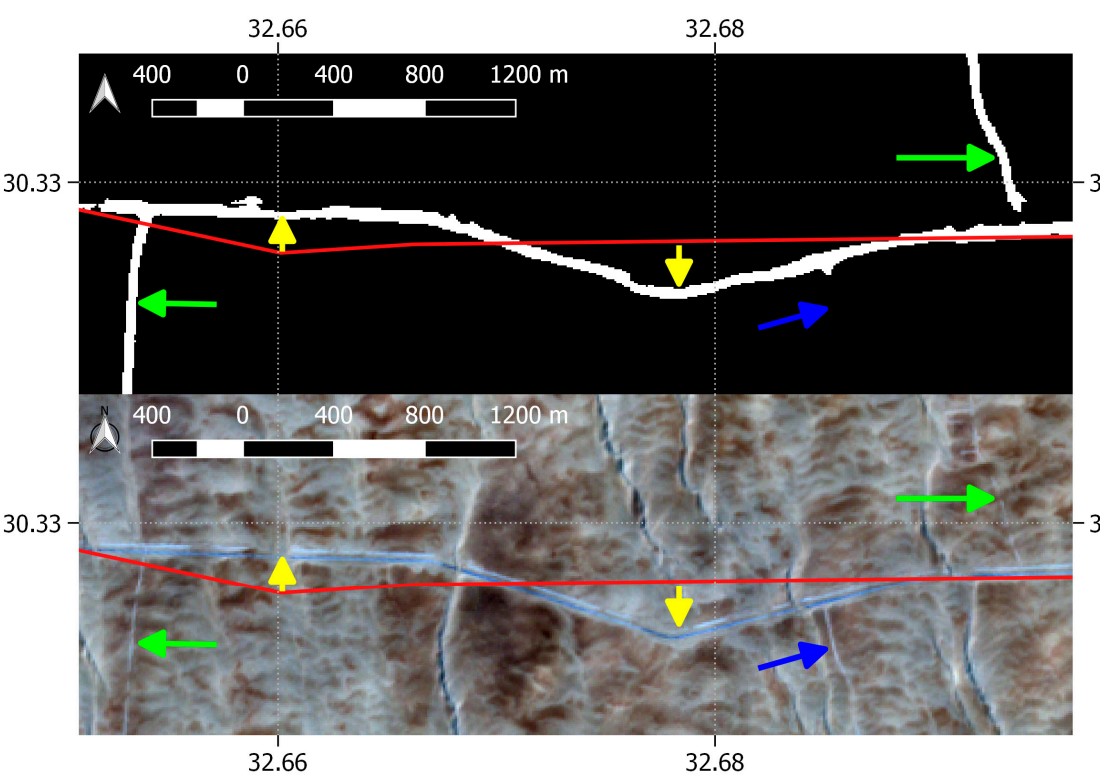

**Figure 5.** Above: mask of detected roads. Below: Sentinel–2 image. Red line shows the Open Street Map (OSM) road data overlaid. The yellow arrows highlight the misregistration between the OSM road and both detected (mask) and actual (Sentinel–2) roads. Green arrows show roads which are not in the OSM dataset. Blue arrows point to a road which was neither in the OSM data, nor detected by the model. Contains modified Copernicus Sentinel–2 data 2020.

Figure 5 demonstrates the success in using, in some cases, low quality OSM data to train the U–Net model (accurate detections despite misregistration of the OSM with actual roads), while also highlighting the various problems with using OSM data as a reference for performance evaluation: the varying width of roads, missing roads in the OSM data, misregistration of the OSM. The top part of this figure shows the mask of detected roads over a part of the North Sinai AOI, while the lower part is a true colour Sentinel–2 image acquired on 2 August 2019 (roughly in the middle of the Sentinel–1 time

series used as an input to the model). White lines in the mask correspond with road detections. In this case the model detected the correct location of the road despite the misregistration of the OSM, which was used to train the model. Roads branching off the main road segment are not in the OSM dataset, but have been detected by the model (apart from one branch which was not detected).

These challenges resulted in the automatically calculated Jaccard index rarely exceeding 0.5 during model training, and the loss function seldom dropping below 0.3. As a relative assessment of performance, this was sufficient for model training and validation. For a more accurate quantitative assessment of results however, this was not sufficient, and a more thorough technique was adopted.

For a more robust accuracy assessment, the following was carried out: For each area, a $0.2 \times 0.2$ degree image subset was randomly selected. To avoid subsets with sparse detections, a threshold was applied to enable only those with at least 7000 road pixels to be considered. In the selected subsets, all roads were manually digitised as vectors, using the Sentinel–1 data and Sentinel–2 data from the same date range as the references. The model detected roads for the same area were similarly digitised. The resulting vector layers were visually inspected and the model detected vector components were assigned labels for true or false positives. All vector layers were converted to raster (one pixel width). A confusion matrix was created by quantifying the true and false positives and negatives. Based on this confusion matrix, the Jaccard index was calculated. Any OSM roads present in the selected subsets were discarded from the analysis since these had been used for training. While this method was suitable for quantifying true or false positives and negatives, another metric was required to assess the positional accuracy of the detections. For this, buffers of a two pixel width (40 m) were first created around both the reference and model detections. The percentage of reference data covered by model detections (completeness) and the percentage of model detections covered by the reference data (correctness) were calculated [15]. From these, the rank distance was derived, using the formula [47]:

$$\text{Rank Distance} \equiv \sqrt{\frac{\%\,\text{Complete}^2 + \%\,\text{Correct}^2}{2}} \tag{3}$$

The two pixel buffer was necessary to account for the varying width of roads and errors in manual digitising, but is a reasonable value when compared to other studies, e.g., [4].

The manually digitised reference subsets in each AOI could have been used to assess the accuracy of the OSM data. However, each validation subset only had a small quantity of OSM roads—in the Taklimakan Desert subset there were none at all (the minimum 7000 road pixel threshold applied only to model detected roads). An assessment of the accuracy of these would not have been representative. Moreover, there have been several dedicated studies on the accuracy of OSM data, e.g., [49–51].

## 3. Results

The model performed well in all areas, despite the diverse landscape forms and road conditions encountered in each. Especially in VV polarisation, many sand dunes produced a high backscatter. This is typical when the incidence angle of the SAR system equals the angle of repose of sand dunes [52]. The model proved nonetheless capable of distinguishing roads from sand dunes, rock formations and other natural features with similar backscatter characteristics as roads.

Of the various SAR input data types, the VV backscatter average alone proved the most effective for both the North Sinai Desert and Grand Erg Oriental. Only for the Taklimakan Desert site did all three layers of coherence, VV backscatter and VH backscatter yield the best results. Table 3 lists the IoU accuracies for each of the single SAR input layers, and for all three input layers for each of the AOIs. The fact that the use of all the input layers improved the results in only one area shows that more information provided to a model is not necessarily better. The decreased accuracy caused by the inclusion of the VH backscatter and coherence is perhaps due to the increase in speckle in these layers. This may have hindered the models in distinguishing particularly challenging roads, such as those that may be narrow, unpaved, or partially buried. The VV backscatter return over this type

of landscape is generally much stronger than the VH backscatter, and enables a clearer distinction of roads. The exception of the Taklimakan desert is perhaps due to the predominant type of road surface and surrounding context of sand dunes. The results for each area are discussed in more detail in the subsections below.

**Table 3.** Intersection over union (IoU) accuracy of road detection models with different SAR input types. The best results for each area are highlighted in bold.

| SAR Input to U–Net Model | IoU Accuracy (in %) | | |
|:---:|:---:|:---:|:---:|
| | North Sinai | Grand Erg Oriental | Taklimakan Desert |
| VV | **89** | **84** | 68 |
| VH | 65 | 57 | 77 |
| Coherence | 64 | 66 | 71 |
| VV + VH + Coherence | 74 | 66 | **89** |

### 3.1. North Sinai Desert

Figure 6 shows roads detected by the model over a part of the North Sinai AOI. The area includes the location of the randomly selected subset (0.2 × 0.2 degree area) in which a more accurate performance evaluation was carried out. This subset is shown in more detail in Figure 7. Figure 8 shows the corresponding area of the SAR layer used as a model input, which was the Sentinel–1 multitemporal backscatter average of the VV polarisation only. Figure 9 shows a Sentinel–2 true colour image of the subset with the available OSM data for this area overlaid. The Sentinel–2 image was acquired on 2 August 2019, which is approximately in the middle of the Sentinel–1 time series (see Table 1).

The confusion matrix for the accuracy assessment is shown in Table 4. Table 5 shows the values of various accuracy indices. The average Jaccard similarity coefficient is 89% and the rank distance is 80%. There were few false positives, i.e., non–roads classified as roads, despite the many natural features of high backscatter that could have been misinterpreted by the model as roads, such as sand dune ridges. However, there were many more false negatives, i.e., undetected road segments. In some cases, the broken segments shown in the mask were correct in that the actual road was partially buried in segments (see Figure 10 and arrows in Figures 7–9).

The VH backscatter over the entire area was much lower, and road features much less clearly defined, particularly those that were unpaved and partially sand covered. This may be the reason why the VH backscatter input degraded the results. The coherence layer highlighted very clearly the roads, but overall had much more speckle, which is possibly why this also reduced the quality of the detection.

**Table 4.** Confusion matrix for true and detected roads, with only the VV backscatter as input, calculated for the same area as in Figures 7–9.

| North Sinai Desert Confusion Matrix | Predicted Roads | Predicted Non–Roads |
|:---|:---|:---|
| True roads | 7671 | 2060 |
| True non–roads | 218 | 4,881,581 |

**Table 5.** Accuracy indices calculated for the North Sinai results.

| IoU Accuracy | Rank Distance | Completeness | Correctness |
|:---:|:---:|:---:|:---:|
| 89% | 80% | 71% | 88% |

**Detected roads for part of North Sinai AOI.**

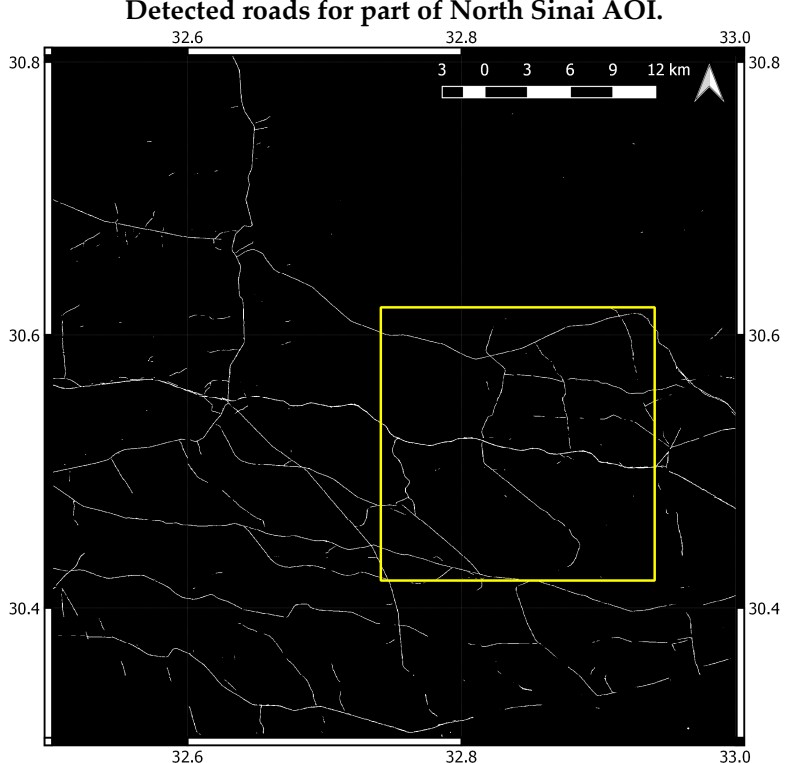

**Figure 6.** Detected roads for part of the North Sinai AOI. The yellow rectangle shows a 0.2 × 0.2 degree subset over which roads were manually digitised and a performance evaluation carried out. This area is shown in more detail in Figure 7.

Detected roads for randomly selected subset of North Sinai AOI

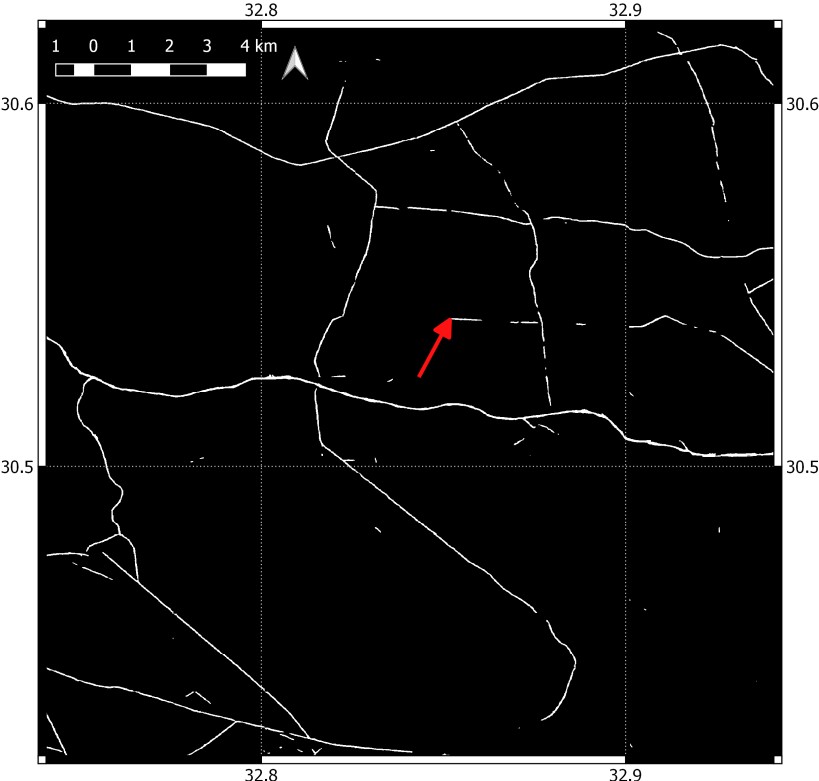

**Figure 7.** Detected roads for a randomly selected 0.2 × 0.2 degree subset over the North Sinai area. White lines correspond with detected roads. The red arrow points to an example of a buried road segment.

Sentinel–1 VV backscsatter average input to North Sinai model

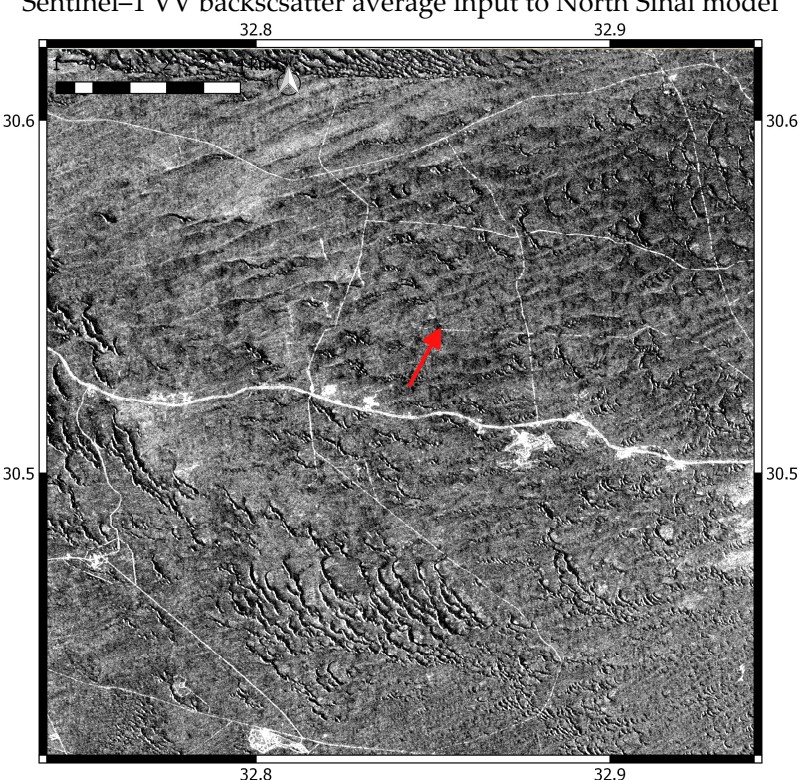

**Figure 8.** The Sentinel–1 average vertical transmit–vertical receive (VV) backscatter used as an input to the model for North Sinai. The area is the same as that of Figure 7. Contains modified Copernicus Sentinel–1 data 2020.

Sentinel–2 image of North Sinai AOI subset

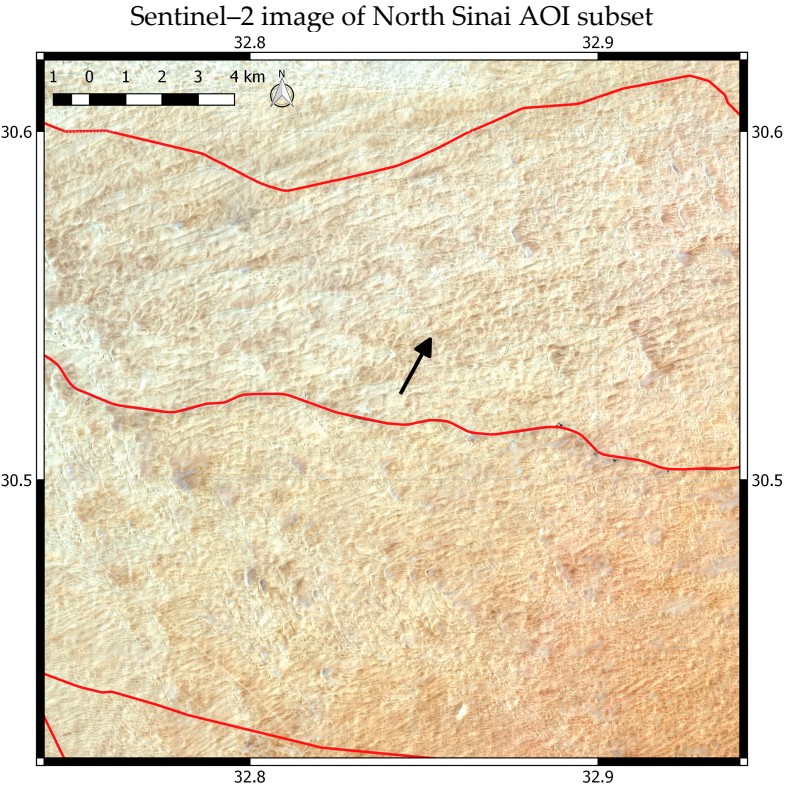

**Figure 9.** Sentinel–2 image of the same area as in Figure 7. The image was acquired on 2 August 2019 (roughly in the middle of the Sentinel–1 time series). It is displayed in true colour, bands 4,3,2 as red,

green, and blue, respectively. Overlaid in red are the available OSM roads for this area. Contains modified Copernicus Sentinel–2 data 2020.

## Detail of partially buried road

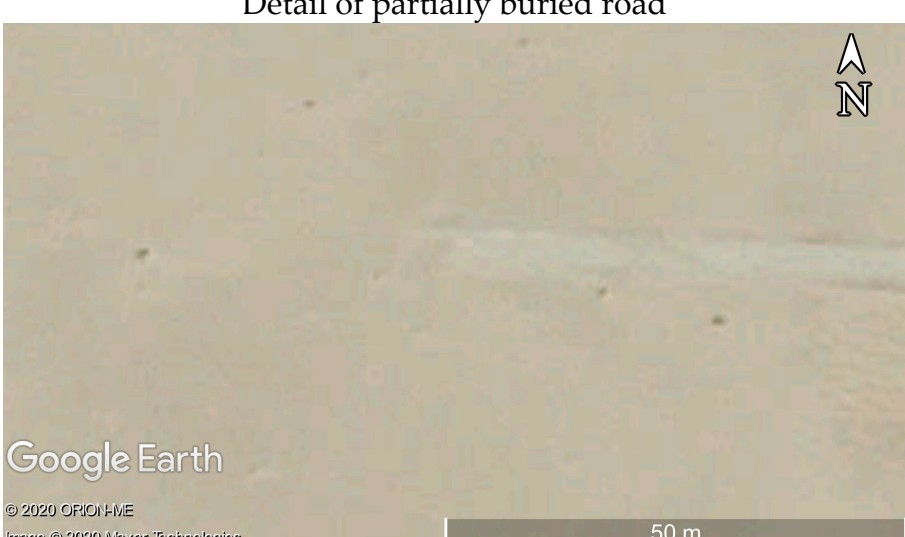

**Figure 10.** Close–up of the buried road segment shown in the very high resolution (VHR) optical data available on Google Earth Pro. The area corresponds with that shown by the red arrow in Figures 7–9. The imagery date is reported to be 5 May 2010.

### 3.2. Grand Erg Oriental

Figure 11 shows the roads detected by the model over a part of the Grand Erg Oriental AOI. The area includes the location of the randomly selected subset (0.2 × 0.2 degree area) in which a more accurate performance evaluation was carried out. This subset is shown in more detail in Figure 12. Figure 13 displays the Sentinel–1 input (VV backscatter) of the same area, and Figure 14 the Sentinel–2 image with the location of OSM roads overlaid. The Sentinel–2 image was acquired on 23 July 2019, which is roughly in the middle of the Sentinel–1 time series (see Table 1).

The confusion matrix for the accuracy assessment is shown in Table 6. Table 7 shows the values of various accuracy indices. The average Jaccard similarity coefficient calculated is 84% and the rank distance is 76%. As with the North Sinai evaluation, there were a few false positives, but many more false negatives. However, the evaluation subset area contains infrastructure in addition to roads. A large segment of missed detections, for example, includes a road running parallel with a large infrastructure installation (see Figure 15). Given the width of the structure, in particular as it appears on the Sentinel–1 data (Figure 13), the model may have misinterpreted it as a natural feature.

As with the North Sinai area, the VH backscatter over the entire area was much lower, with road features less clearly defined, while the coherence layer had greater speckle. These may be the reasons why the results were better with the VV backscatter alone.

**Table 6.** Confusion matrix for true and detected roads calculated for same area as in Figures 12–14.

| Grand Erg Oriental Confusion Matrix | Predicted Roads | Predicted Non–Roads |
|:---:|:---:|:---:|
| **True roads** | 14,692 | 6430 |
| **True non–roads** | 282 | 4,929,956 |

**Table 7.** Accuracy indices calculated for the Grand Erg Oriental results.

| IoU Accuracy | Rank Distance | Completeness | Correctness |
|:---:|:---:|:---:|:---:|
| 84% | 76% | 64% | 86% |

Detected roads for part of Grand Erg Oriental AOI

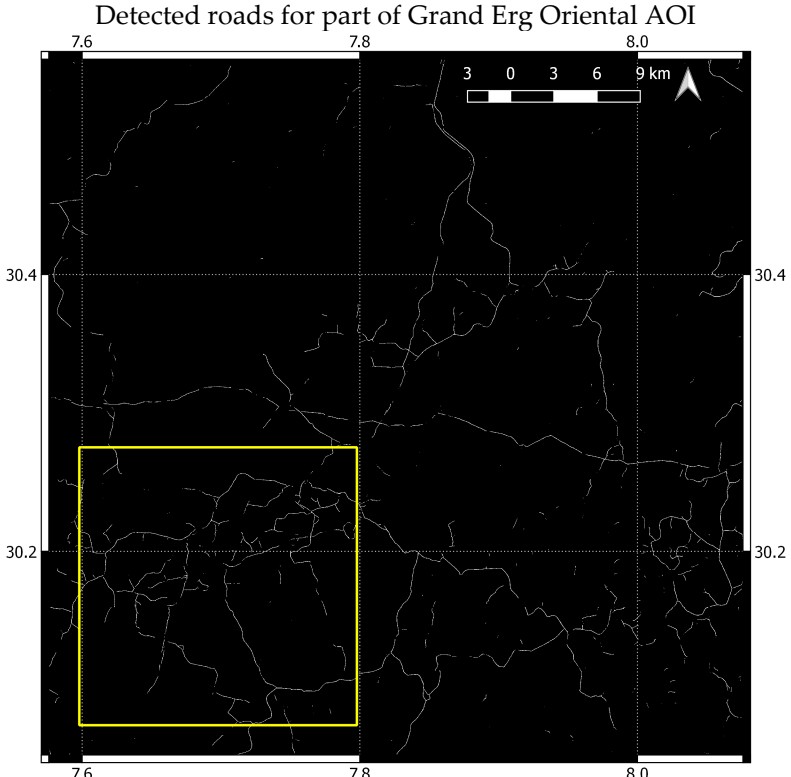

**Figure 11.** Detected roads for part of the Grand Erg Oriental AOI. The yellow rectangle shows a 0.2 × 0.2 degree subset over which roads were manually digitised and a performance evaluation carried out. This area is shown in more detail in Figure 12.

Detected roads for randomly selected subset of Grand Erg Oriental AOI

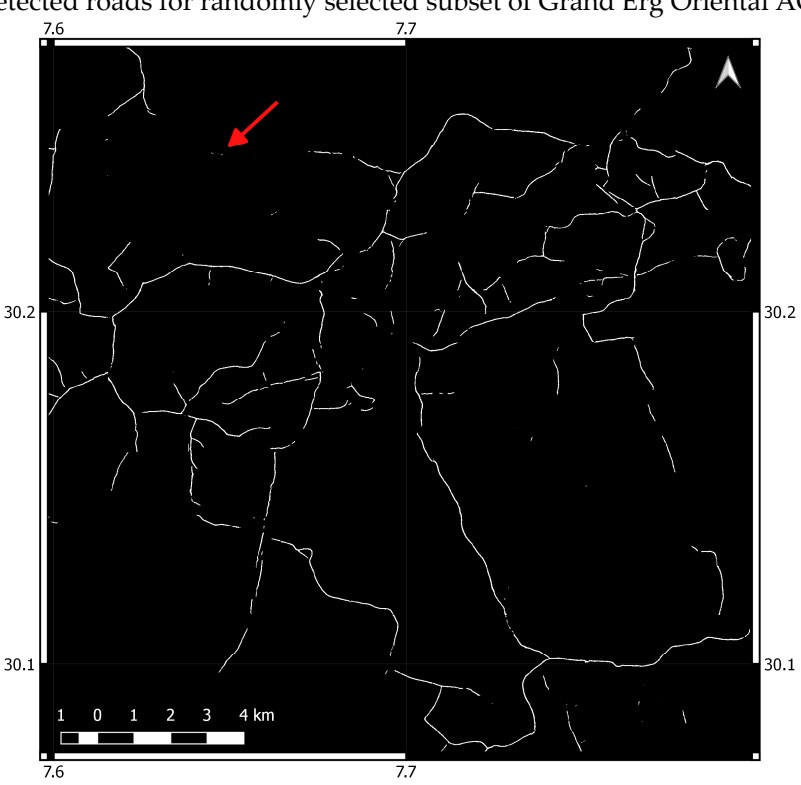

**Figure 12.** Detected roads for a randomly selected 0.2 × 0.2 degree subset over the Grand Erg Oriental AOI. White lines correspond with detected roads. The red arrow points to an example of a missed detection, perhaps due to mixed infrastructure.

Sentinel–1 VV backscsatter average input to Grand Erg Oriental model

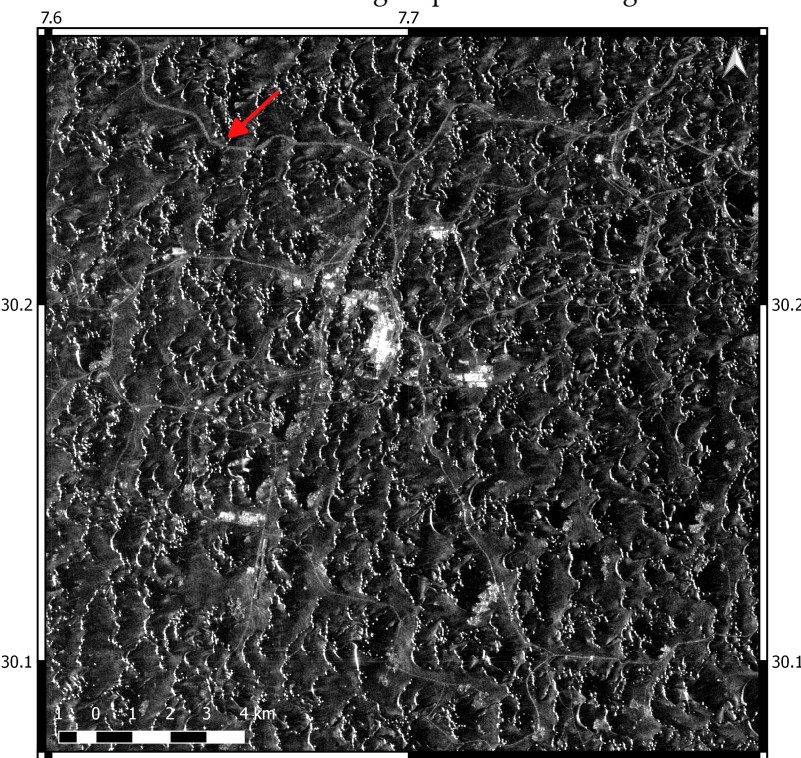

**Figure 13.** The Sentinel–1 average VV backscatter used as an input to the model for the Grand Erg Oriental. The area is the same as that of Figure 12. Contains modified Copernicus Sentinel–1 data 2020.

Sentinel–2 image of Grand Erg Oriental AOI subset

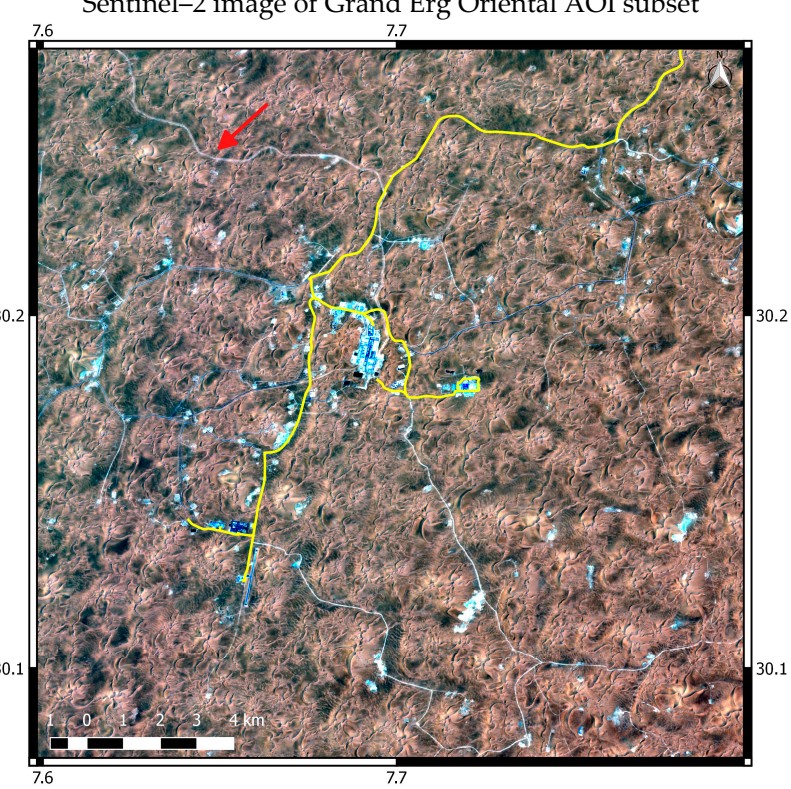

**Figure 14.** Sentinel–2 image of the same area as in Figure 12. The image was acquired on 23 July 2019 (roughly in the middle of the Sentinel–1 time series). It is displayed in true colour, bands 4,3,2 as red,

green, and blue, respectively. Overlaid in yellow are the available OSM roads for this area. Contains modified Copernicus Sentinel data 2020.

### Detail of mixed infrastructure

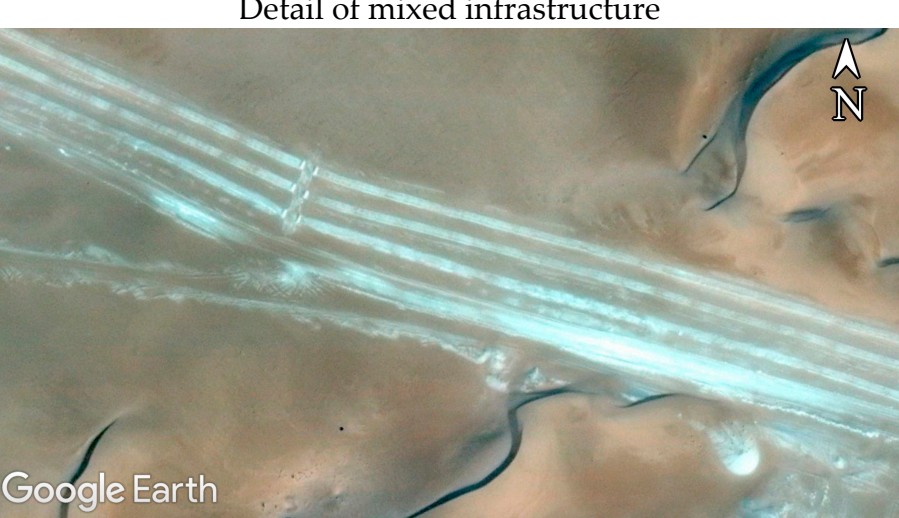

**Figure 15.** Close–up of a road segment in VHR optical data available on Google Earth Pro. The area corresponds with that shown by the red arrow in Figures 12–14. The road segment was not detected by the model. As evident in the figure, the road runs parallel with other infrastructure, which may have affected the ability of the model to correctly interpret the scene. The imagery date is reported to be 13 September 2013.

### 3.3. Taklimakan Desert

Figure 16 shows roads detected by the model over a part of the Taklimakan Desert AOI. The area includes the location of the randomly selected subset (0.2 × 0.2 degree area) in which a more accurate performance evaluation was carried out. This subset is shown in more detail in Figure 17. Figure 18 shows the Sentinel–1 input as a red, green, and blue combination of VH and VV backscatter and coherence averages, respectively. Figure 19 shows a Sentinel–2 image of the same area. In this subset there were no OSM roads. The Sentinel–2 image was acquired on 31 July 2019, approximately in the middle of the Sentinel–1 time series (see Table 1).

The confusion matrix for the accuracy assessment is shown in Table 8. Table 9 shows the values of various accuracy indices. The average Jaccard similarity coefficient calculated is 89% and the rank distance is 75%. As with the other areas, there were more false negatives than false positives. Compared to the other areas, there appear to be more paved roads, with straighter paths. The sand dunes are larger and do not have the characteristic lines of high backscatter apparent in the other two areas, although some sparse misclassifications still arise over natural features.

This was the only area where the best results were obtained with all three SAR input layers of coherence, VV backscatter and VH backscatter. The backscatter over sand dunes is much lower in VH than VV, while the road features are still clearly defined, perhaps due to the high relative permittivity of the paved roads. This may be the added value of the VH layer. The coherence layer still displayed much speckle over the sand dunes, but the roads were very clearly defined, perhaps again due to the material of their construction.

Sand drift encroachment on roads in the Taklimakan desert is a serious problem and many efforts have been made to mitigate the issue [7,53,54]. Figure 20 shows a road segment of the subset in VHR optical data available on Google Earth Pro, the date of which is reported to be 26 October 2014. The road is partially buried in this image, but the model output shows a continuous, unbroken line. It would seem that maintenance had been carried out on this road in between the date of the VHR optical image acquisition and the date range of the Sentinel–1 time series used as an input to the model.

Detected roads for part of Taklimakan Desert AOI

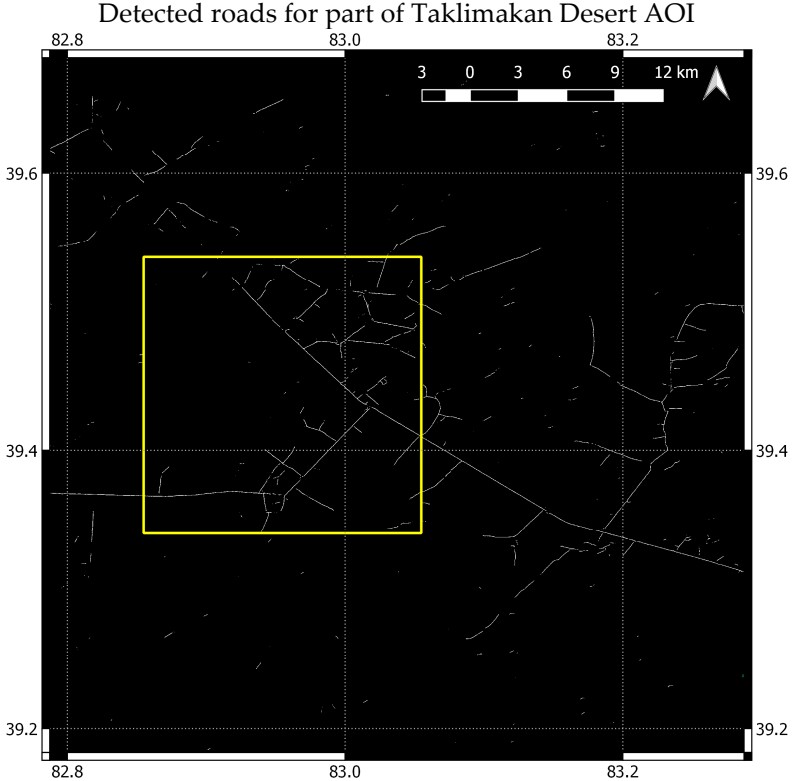

**Figure 16.** Detected roads for part of the Taklimakan Desert AOI. The yellow rectangle shows a 0.2 × 0.2 degree subset over which roads were manually digitised and a performance evaluation carried out. This area is shown in more detail in Figure 17.

Detected roads for randomly selected subset of Taklimakan Desert AOI

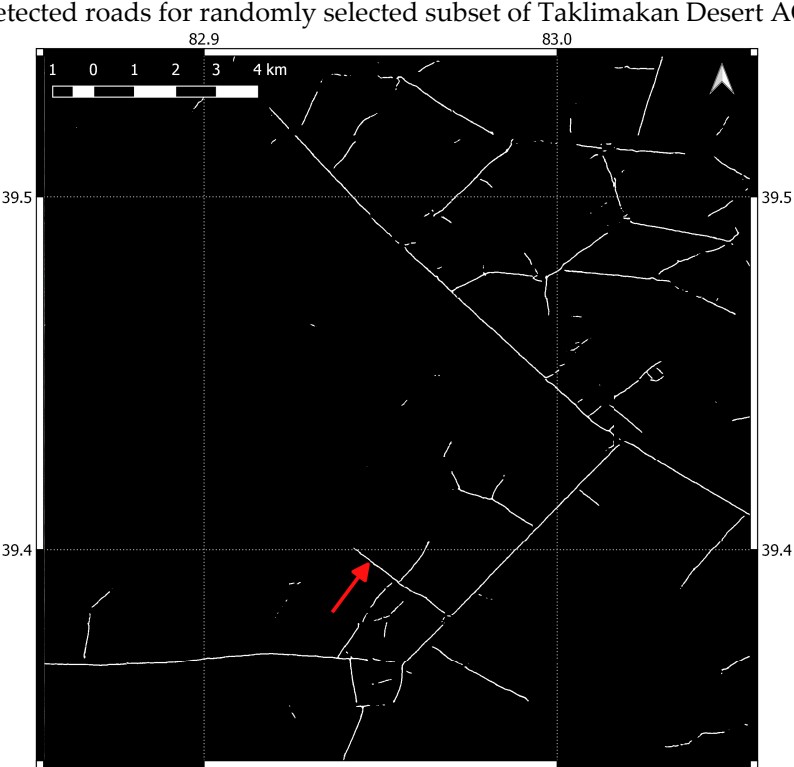

**Figure 17.** Detected roads for a randomly selected 0.2 × 0.2 degree subset over the Taklimakan Desert AOI. White lines correspond with detected roads. The red arrow points to an example of a formerly buried road segment.

Sentinel–1 VH, VV and Coherence input to Taklimakan Desert model

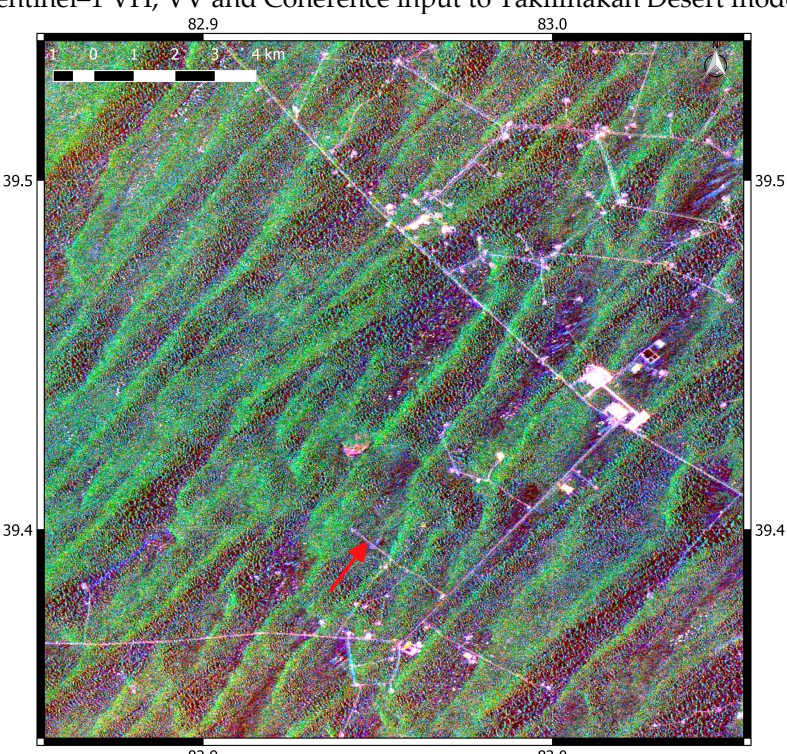

**Figure 18.** Sentinel–1 colour composite of time series averages of the VH backscatter in red, VV backscatter in green, and coherence in blue. This comprised the input to the model for the Taklimakan Desert area. The extent is the same as that for Figure 17. Contains modified Copernicus Sentinel data 2020.

Sentinel–2 image of Taklimakan Desert AOI subset

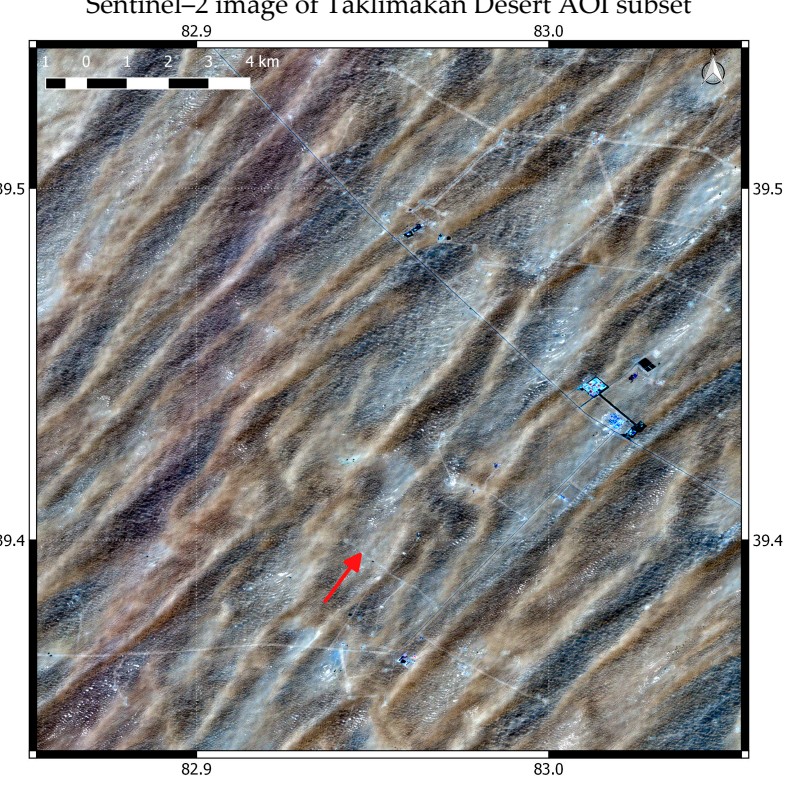

**Figure 19.** Sentinel–2 image of the same area as in Figure 17. The image was acquired on 31 July 2019 (roughly in the middle of the Sentinel–1 time series). It is displayed in true colour, bands 4,3,2 as red, green, and blue, respectively. Contains modified Copernicus Sentinel data 2020.

**Table 8.** Confusion matrix for true and detected roads calculated for the same area as in Figures 17–19.

| Taklimakan Desert Confusion Matrix | Predicted Roads | Predicted Non–Roads |
| --- | --- | --- |
| **True roads** | 11,077 | 2622 |
| **True non–roads** | 428 | 4,940,949 |

**Table 9.** Accuracy metrics calculated for the Taklimakan Desert results.

| IoU Accuracy | Rank Distance | Completeness | Correctness |
| --- | --- | --- | --- |
| 89% | 75% | 69% | 81% |

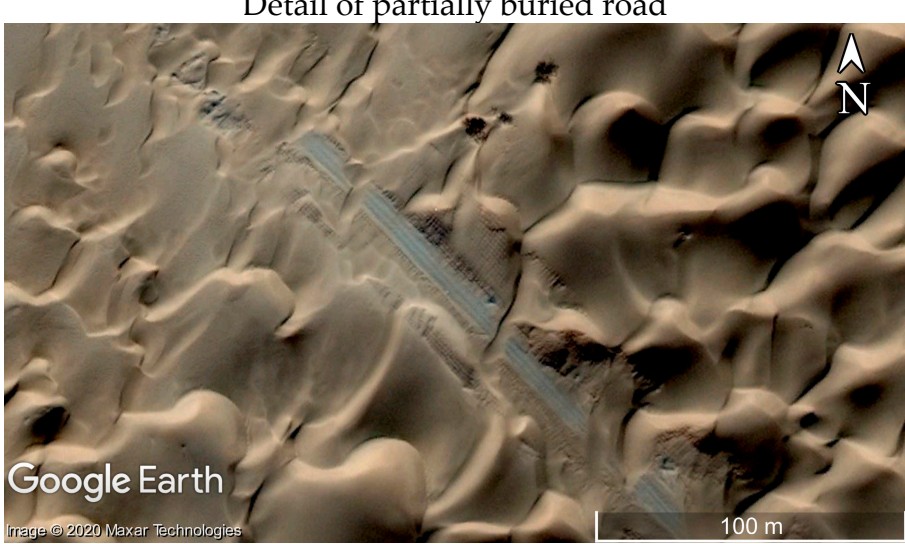

Detail of partially buried road

**Figure 20.** Close–up of a road segment in VHR optical data available on Google Earth Pro. The area corresponds with that shown by the red arrow in Figures 17–19. The road appears to be partially buried, while the output of the model shows a continuous line. It would appear the road was cleared some time between the acquisition of this image and the date range of the Sentinel–1 time series used as the model input. The imagery date is reported to be 26 October 2014.

## 4. Discussion

The road detection methodology proposed here aims above all to demonstrate the potential for low cost, but reliable mapping and monitoring of road networks, that can easily be adapted and transferred to extensive regions. It has been applied to three desert areas, each covering around 47,500 km2, each corresponding to the footprint of one Sentinel–1 IW scene. The results over all three areas have achieved an IoU accuracy of over 80%. This accuracy metric takes into account the class imbalance, which is typically the case for road detection in satellite imagery, where in any given area the non–road pixels are expected to greatly outnumber the road pixels. The rank distance is over 75% in all the areas tested, which demonstrates the proximity (in completeness and correctness space) between the reference and model detected roads. The recent study of Abdelfattah and Chokmani [4], who also used Sentinel–1 for road detection over a similar area, could be considered a benchmark to assess the performance of the methodology described here. The correctness and completeness of detections reported in their study are at least 10 percentage points below those described in this paper, for each of the AOIs, and despite the fact that the buffers surrounding the reference objects and detections were larger (three pixels as opposed to two pixels in the study reported here). However, the authors did focus on smaller roads and tracks, while the research of this paper includes major roads in addition to smaller unpaved tracks, so this comparison needs to be treated with caution.

Each desert area in which the methodology was tested has different characteristics, such as diverse sand dune forms, varying predominant road types, and the presence of other infrastructure. Despite these differences, the algorithm performs well and therefore demonstrates robustness. Moreover, all the required input is free and open, including the satellite data from which roads are detected, and the OSM reference data. The methodology is hence cost–effective. To scale to other areas, the model does require training. Attempts to apply pre–trained models to the other AOIs resulted in a poor performance, probably due in large part to the greatly varying sand dune morphology in each area. However, the U–Net model architecture is particularly efficient, and training with one NVIDIA GTX1080 Ti GPU never took more than around 25 min. The capability of the algorithm to work well over multiple areas with the available OSM data, without manual intervention for reference dataset cleaning, demonstrates scalability.

Despite the success of the model there are some limitations. While there were few false positives, a significantly higher number of false negatives were encountered, where the algorithm failed to detect roads. This was consistent across all test areas. Many of these false positives were due to the complexity of the context in which the road was situated. Some roads, for example, ran alongside other infrastructure, which in the resolution cell of the SAR input data could be misinterpreted as natural or other non–road features. A possible solution to mitigate these missed detections could be to expand the training set to include more OSM training samples over a wider area, or to include additional classes with a mixed infrastructure.

Another limitation is related to the required input data. In order to reduce speckle while preserving the spatial resolution, speckle filtering was carried out in the temporal domain. This requires processing of a time series, which is computationally expensive, especially for the average coherence generation with SLC format data. However, in most cases it was demonstrated that the VV average backscatter alone produces the best results. Only in one case did the coherence improve the results of the detection, but not significantly. The added value of the coherence is perhaps in those cases where there is a high proportion of roads made from material characterised by a high coherence, i.e., those that have a stable surface, such as paved roads. These would contrast highly with the surrounding sand, where volume decorrelation causes low coherence. In cases where roads are unpaved, or partially sand covered, the coherence is perhaps too noisy, despite the multitemporal averaging, and may degrade the results. Particularly in less developed areas, where there may be fewer paved roads, the coherence processing could therefore be discarded in the interest of a more computationally efficient algorithm.

The VH backscatter over all sites was much weaker than the VV. As with the coherence, the VH backscatter input only improved the detection results over the Taklimakan Desert site. Here, the VH backscatter was less pronounced over sand dunes, while still high enough over roads to enable their distinction. This may have helped reduce the ambiguity between the high backscatter encountered in the VV polarisation at certain sand dune inclinations with roads. Again, the unique suitability of the VH channel in this area alone may be due to the high relative permittivity of the roads, causing a high enough backscatter even in the weak cross polarisation channel. Elsewhere, however, the low backscatter return of the VH over less reflective roads may have contributed to the degradation of the results.

In terms of the utility of the algorithm for operational road detection in desert areas, the low number of false positives are advantageous in any alert system. Committing resources to detect human activity in remote areas is expensive and time consuming, especially in developing countries such as in North Africa where the means for such activities may be limited. As a monitoring system, the chosen time series length constrains the maximum frequency of monitoring to at least two and a half months. It may be possible to reduce this and still achieve good results. However, any changes significant enough to be observed at the spatial scale of the model are unlikely to occur at temporal frequencies significantly higher than this.

The algorithm proposed here is only a prototype. Improvements could be made for example to reduce the number of missed detections in challenging areas, perhaps by expanding the training set,

and including other classes. It could be interesting also to assess the extent to which the Sentinel–1 stripmap (SM) mode may improve detections. While the higher resolution SM mode (around $10 \times 10$ m in GRD format) is not as systematically available as the Sentinel–1 IW mode, it may nonetheless be useful to detect smaller roads and tracks that may not be resolved in IW mode. This is particularly relevant for security applications [4]. However, the methodology applied with Sentinel–1 IW data nonetheless demonstrates the potential for regular and large–scale mapping and monitoring of desert roads.

## 5. Conclusions

The methodology proposed here for road detection in desert areas, using Sentinel–1 SAR data as an input and OSM data for training, has the potential to provide a robust, cost–effective and scalable solution for the mapping and monitoring of road networks in desert areas. This methodology is still a prototype that has been tested in three areas, each the size of one Sentinel–1 IW scene. More work is required to test its performance over a wider area and over different desert landscape types. Possible improvements with the Sentinel–1 SM mode could be explored. While the accuracy assessments over the AOIs resulted in Jaccard similarity coefficients above 84% and rank distances of over 75%, more work still needs to be done to improve the accuracy, in particular to reduce the number of missed detections. Future improvements may include the addition of other infrastructure classes, or mixed classes, to account for roads in the proximity of other structures. The methodology may be further tested to quantify model improvement according to the quantity of training data. Additionally, more experimentation can be carried out with additional data augmentation techniques, such as those that modify the intensity of pixels, rather than their spatial position alone. More importantly, the utility of the system needs to be tested by real end users. Its success should be measured against the available systems already in place. Such pre–existing systems are likely to vary between different users and geographic regions. Any improvements should be tailored to meet specific user requirements. The objective of the work presented here is to assess the benefits of EO and open data in combination with deep learning for cost–effective and large–scale monitoring. The ambition is to ultimately improve operational road detection and monitoring to support decision–making. With an increasing global population, dynamic migration patterns, and with expanding and evolving road networks, the need for efficient monitoring systems is ever more critical.

**Author Contributions:** Conceptualization, C.S.; Data curation, C.S., M.L. and A.L.; Formal analysis, C.S.; Investigation, C.S.; Methodology, C.S.; Project administration, C.S. and S.A.; Software, C.S.; Supervision, M.L. and A.L.; Validation, C.S.; Writing—original draft, C.S.; Writing—review & editing, M.L. and A.L. All authors have read and agreed to the published version of the manuscript.

**Funding:** This research received no external funding.

**Acknowledgments:** The authors would like to thank the following for their contribution to this research: Jerin Paul for the modified U–Net model that was adopted in this methodology [36]; the Copernicus programme, which provides free and open access to Sentinel data; Open Street Map, for the free provision of vector data of roads, and Lucio Colaiacomo (SatCen) for the preliminary preparation of this OSM data; CreoDIAS for access to the cloud processing environment for Sentinel–1 data processing; and the Advanced Concepts Team of ESA, for access to Sandy, the GPU sandbox environment.

**Conflicts of Interest:** The authors declare no conflict of interest.

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
