# Peer review of "Deep Learning with Open Data for Desert Road Mapping"

_remotesensing, doi:10.3390/rs12142274_

Round 1

Reviewer 1 Report

This study provides a whole deep learning framework for desert road detection in SAR satellite data, including the data preprocess procedure and the details of the deep learning. On extensive data, the proposed shows promising results. However, the following problems should be well addressed before acceptance.

1. In Figure 4, the original image is different from its augmented versions. It seems that the original image only contains partial scene in the augmented versions.

2. In the manuscript, many references are corrupted, e.g., line 339, 371, 378, 381, 383, etc., which makes it difficult to read.

3. In Table 3, “VV+ VH + Coherence” contains much more input information than others, however, it only obtain the best performance in the Taklimakan Desert data. Please explain this in details.

4. It is unclear how to compute the confusion matrix in Table 4,5. How to determine a road is predicted?

5. To demonstrate the effectiveness of the deep learning method, it is necessary to compare the proposed method with some other existing baselines.

6. To make introduction for CNNs complete, some recent CNNs based works[1-3] should be supplemented into the introduction.

[1] Adaptive importance learning for improving lightweight image super-resolution network. 2019

[2] Ace: Adapting to changing environments for semantic segmentation. 2019.

[3] Class-Based Styling: Real-time Localized Style Transfer with Semantic Segmentation. 2019

Author Response

  1. In Figure 4, the original image is different from its augmented versions. It seems that the original image only contains partial scene in the augmented versions.

The augmented versions take the exact same scene from the original. Following rotation, gaps in the corners are filled using “reflect” mode, which takes the mirror image of adjacent pixels and thus avoids broken segments. This is now better explained in the caption of Figure 4.

  1. In the manuscript, many references are corrupted, e.g., line 339, 371, 378, 381, 383, etc., which makes it difficult to read.

This has now been fixed.

  1. In Table 3, “VV+ VH + Coherence” contains much more input information than others, however, it only obtain the best performance in the Taklimakan Desert data. Please explain this in details.

This is now explained in lines 415 to 423.

  1. It is unclear how to compute the confusion matrix in Table 4,5. How to determine a road is predicted?

The confusion matrix of true and false positives and negatives was created by counting the pixels corresponding to each in the model detections and manually digitised roads. Based on this confusion matrix, the Jaccard index was calculated. This is now described in lines 383 to 386.

An additional assessment method that considers also the proximity between predictions and reference locations has been carried out and reported in the paper in lines 386 to 397. This method includes calculation of the Rank Distance, and associated Completeness (percentage of reference data covered by model detections) and Correctness (percentage of model detections covered by reference data).

  1. To demonstrate the effectiveness of the deep learning method, it is necessary to compare the proposed method with some other existing baselines.

A comparison has now been made to a similar study, carried out by Abdelfattah and Chokmani [ref 4], who also use Sentinel-1 for automatic road detection in similar areas, using similar accuracy metrics (additional validation has now been carried out using Rank Distance, Percentage Completeness and Correctness). The figures for Percentage Completeness and Correctness reported by them are worse by at least 10 percentage points than those of our study. However, their study focuses only on small roads and tracks over a limited area, so this comparison needs to be treated with caution. This is explained in lines 529 to 537.

  1. To make introduction for CNNs complete, some recent CNNs based works[1-3] should be supplemented into the introduction.

[1] Adaptive importance learning for improving lightweight image super-resolution network. 2019

[2] Ace: Adapting to changing environments for semantic segmentation. 2019.

[3] Class-Based Styling: Real-time Localized Style Transfer with Semantic Segmentation. 2019

An updated literature review on CNNs has been provided in the introduction, with many more references, including 1 and 2 from the above. Please see modified text in lines 48 to 66.

Reviewer 2 Report

The paper presents cost-effective and large scale monitoring whose value arises via a combination of available open sources such as EO data, SAR data, and AI.

The result looks good, but are there any benchmarkable external references? For example, ref [3] based on the SAR imagery along the Tunisian-Libyan border. It would be the gest if the proposed method outperforms ref[3], but even if it is not the case, the proposed approach has a relative advantage in terms of full automation.

Author Response

The result looks good, but are there any benchmarkable external references? For example, ref [3] based on the SAR imagery along the Tunisian-Libyan border. It would be the gest if the proposed method outperforms ref[3], but even if it is not the case, the proposed approach has a relative advantage in terms of full automation.

A comparison has now been made with the study of Abdelfattah and Chokmani, (now ref[4]), which is the closest to this study, given they also use Sentinel-1 for automatic road detection in similar areas, using similar accuracy metrics (additional validation has now been carried out using Rank Distance, Percentage Completeness and Correctness). The figures for Percentage Completeness and Correctness reported by them are worse by at least 10 percentage points than those of our study. However, their study focuses only on small roads and tracks over a limited area, so this comparison needs to be treated with caution. This is explained in lines 529 to 537.

Reviewer 3 Report

  1. It was not clear to me whether your models were initiated using pre-trained weights (such as those used by Paul) or whether they were initiated with random weights. Can you clarify this?
  2. A more detailed discussion and explanation of UNets and semantic segmentation methods in general should be provided in the Background section. 
  3. Given that the goal here is to extract linear features, an assessment metric that compares similarity between reference linear features and the predicted linear features that takes into account the similarity between the features and how far they are apart would be informative. The pixel-based assessment methods seem flawed in this case since the divergence distance between predictions and reference locations is not incorporated. Prior to a revision, I would recommend that a more robust assessment be incorporated that is more appropriate for line features. Here are some references to investigate:A Review of Techniques for Extracting Linear Features from Imagery by Quackenbush and Review of Automatic Feature Extraction from High-Resolution Optical Sensor Data for UAV-Based
    Cadastral Mapping by Crommelinck et al.
  4. Given that you generated some manually digitized reference data to assess the models, would it also be possible to assess the OSM data quality against the same reference? Given that a large component of this study relates to using imperfect reference data, quantifying the error in the reference data would be useful. 
  5. I would be interested in how the different models perform when used to predict the other sites. Even if the results are poor, it would be interesting to comment on how well the models generalize or transfer to other regions, as this is commonly sited as a strength of deep learning. 
  6. Can you more clearly explain how the UNet hyperparameters were altered in comparison to the model created by Paul? What augmentations did you make to make this specific UNet implementation more suitable for your study?
  7. The PDF that I reviewed had some issues with missing citations (Reference Link Missing) that needs fixed. Maybe this was an issue with your citation manager. 

Author Response

Please see attached "Reviewer3.docx"

Round 2

Reviewer 1 Report

All my concerns have bee well addressed. The current version is ready for publication.

Reviewer 3 Report

The authors have addressed my prior concerns/comments. I have not further suggestions and recommend publication.